# Fine-tuning of substrate preferences of the Src-family kinase Lck revealed through a high-throughput specificity screen

**Neel H Shah[1,2,3], Mark Löbel[1,2,3], Arthur Weiss[4,5], John Kuriyan[1,2,3,6]\***

[1]Department of Molecular and Cell Biology, University of California, Berkeley, Berkeley, United States; [2]California Institute for Quantitative Biosciences, University of California, Berkeley, Berkeley, United States; [3]Howard Hughes Medical Institute, University of California, Berkeley, Berkeley, United States; [4]Department of Medicine, Rosalind Russell/Ephraim P Engleman Rheumatology Research Center, University of California, San Francisco, San Francisco, United States; [5]Howard Hughes Medical Institute, University of California, San Francisco, San Francisco, United States; [6]Molecular Biophysics and Integrated Bioimaging Division, Lawrence Berkeley National Laboratory, Berkeley, United States

**\*For correspondence:**
kuriyan@berkeley.edu

**Abstract** The specificity of tyrosine kinases is attributed predominantly to localization effects dictated by non-catalytic domains. We developed a method to profile the specificities of tyrosine kinases by combining bacterial surface-display of peptide libraries with next-generation sequencing. Using this, we showed that the tyrosine kinase ZAP-70, which is critical for T cell signaling, discriminates substrates through an electrostatic selection mechanism encoded within its catalytic domain (Shah et al., 2016). Here, we expand this high-throughput platform to analyze the intrinsic specificity of any tyrosine kinase domain against thousands of peptides derived from human tyrosine phosphorylation sites. Using this approach, we find a difference in the electrostatic recognition of substrates between the closely related Src-family kinases Lck and c-Src. This divergence likely reflects the specialization of Lck to act in concert with ZAP-70 in T cell signaling. These results point to the importance of direct recognition at the kinase active site in fine-tuning specificity.

DOI: https://doi.org/10.7554/eLife.35190.001

## Introduction

The ~30 cytoplasmic tyrosine kinases are characterized by the presence of a tyrosine kinase domain and one or more modular binding domains, such as SH2 and SH3 domains (*Robinson et al., 2000*). The catalytic activities of the tyrosine kinase domains are weak and, as a consequence, the binding domains strongly influence specificity by localizing the kinases to the vicinity of phosphorylation targets (*Miller, 2003*). The individual catalytic domains of tyrosine kinases display some intrinsic preference for sequence features surrounding the phosphorylated tyrosine residues in their substrates (*Songyang et al., 1995*), but the current understanding of tyrosine kinase mechanism emphasizes localization by the modular binding domains as a principal mechanism for gaining specificity.

We have shown recently that there is an important role for direct substrate selection by the kinase active sites of two cytoplasmic tyrosine kinases, the Src-family kinase Lck and the Syk-family kinase ZAP-70, in the response of T cells to antigen recognition (*Shah et al., 2016*). T cell activation requires the sequential action of Lck and ZAP-70 (*Figure 1A*) (*Iwashima et al., 1994*; *Weiss and*

*Littman, 1994*; *Chakraborty and Weiss, 2014*). First, Lck phosphorylates the ITAM motifs of the T cell antigen receptor, creating binding sites for ZAP-70, and Lck then phosphorylates ZAP-70 and releases its autoinhibition. Once recruited to the T cell receptor and activated by Lck, ZAP-70 phosphorylates the scaffold proteins LAT and SLP-76, leading to the recruitment of other signaling proteins, such as phospholipase Cγ1 and the Ras activator SOS via the adapter protein Grb2. The strictly sequential action of Lck and ZAP-70 reflects the inability of the kinase domain of Lck to phosphorylate LAT and SLP-76 efficiently, and the inability of ZAP-70 to phosphorylate ITAMs and to activate itself. This orthogonality in the specificities of the kinase domains of Lck and ZAP-70, combined with the targeting functions of their binding domains, is postulated to underlie the sensitivity and selectivity of the T cell response to antigens.

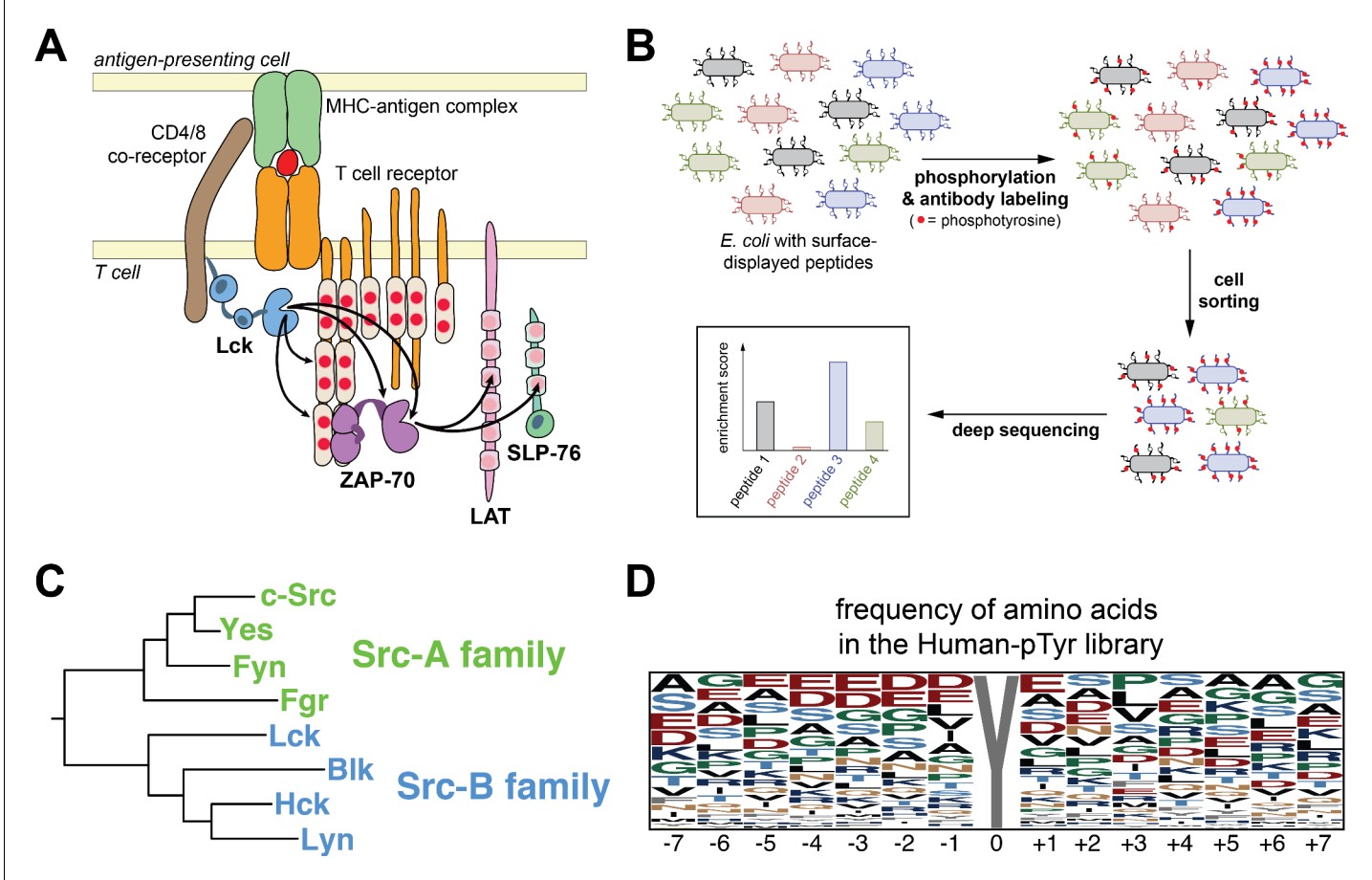

**Figure 1.** Analysis of Src-family kinase specificity using a high-throughput platform. (**A**) T cell receptor-proximal signaling mediated by the tyrosine kinases Lck and ZAP-70 (based on *Au-Yeung et al., 2009*). (**B**) Schematic representation of the high-throughput specificity screening platform. (**C**) Phylogenetic tree inferred from the sequences the eight human Src-family kinase domains, highlighting the segregation of Src-A and Src-B kinases. (**D**) The frequency of amino acid residues at each position in the full Human-pTyr library, visualized using WebLogo (*Crooks et al., 2004*).
DOI: https://doi.org/10.7554/eLife.35190.002

The following figure supplements are available for figure 1:

**Figure supplement 1.** pLogo diagram depicting the probabilities of amino acid residues at each position surrounding the target tyrosine residue in sequences in the Human-pTyr library.
DOI: https://doi.org/10.7554/eLife.35190.003

**Figure supplement 2.** Correlations between enrichment scores for all peptides in the Human-pTyr library between replicate screens performed using the same kinase.
DOI: https://doi.org/10.7554/eLife.35190.004

To analyze the differential specificities of Lck and ZAP-70, we had adapted a high-throughput platform to simultaneously measure the efficiency of phosphorylation by a tyrosine kinase of hundreds of genetically encoded peptides in bacterial surface-display libraries (*Figure 1B*) (*Shah et al., 2016*). In this method, peptides are expressed on the extracellular surface of bacterial cells as fusions to a scaffold protein, with one peptide of defined sequence per cell (*Rice and Daugherty, 2008*). Addition of a purified kinase to the cell suspension results in phosphorylation of the displayed peptides (*Henriques et al., 2013*). The cells are then sorted based on phosphorylation level, as detected by a pan-phosphotyrosine antibody. Deep sequencing of the peptide-coding DNA before and after sorting yields a quantitative metric of the efficiency with which each substrate is phosphorylated by the kinase, the 'enrichment score' (*Shah et al., 2016*). The enrichment score correlates strongly with the rate of phosphorylation measured in vitro using purified peptides and kinases.

Using this high-throughput technique, we had analyzed scanning-mutagenesis libraries of peptides derived from LAT phosphosites and ITAM motifs, and had identified mutations in these peptides that impact their phosphorylation by Lck or ZAP-70 (*Shah et al., 2016*). This analysis showed that ZAP-70 and Lck differentiate their respective substrates on the basis of an electrostatic-selection mechanism that is encoded within their kinase domains. The phosphosites in LAT are flanked by many negatively charged residues, and this increases the efficiency of phosphorylation by ZAP-70 and reduces that by Lck. By contrast, the ITAM motifs on the T cell receptor bear positively-charged residues that deter phosphorylation by ZAP-70, but promote phosphorylation by Lck.

Lck is one of eight Src-family kinases in the human genome (*Figure 1C*). This family of kinases emerged at the very root of the metazoan evolutionary tree, with a clearly identifiable Src-family kinase present in the genomes of choanoflagellates, the closest identified relatives of the true metazoans (*Richter and King, 2013*), and in even more distantly related taxa (*Suga et al., 2012*; *Suga et al., 2014*). The evolution of the Src-family kinases can be traced through duplications and specialization into the present-day members of the family (*Figure 1C*). In humans, the Src-family kinases fall into two groups, the Src-A and Src-B subfamilies (*Hughes, 1996*; *Suga et al., 1997*). The Src-A kinases, which include the extremely well studied kinase c-Src, are expressed in a wide variety of cell types. In contrast, Lck and other Src-B kinases are restricted to cells from the hematopoietic lineage, such as T cells and B cells (*Lowell and Soriano, 1996*; *Thomas and Brugge, 1997*; *Bolen and Brugge, 1997*). Except for unique N-terminal segments that include membrane-localization motifs (*Resh, 1994*), each of the Src-family kinases are closely related in sequence, with 64–90% sequence identity in their catalytic domains, and have very similar three-dimensional structures (*Sicheri et al., 1997*; *Xu et al., 1997*; *Yamaguchi and Hendrickson, 1996*; *Williams et al., 1997*; *Boggon and Eck, 2004*). Prior studies employing peptide library screens with degenerate sequences have shown that all eight of the human Src-family kinases have similar substrate sequence preferences (*Songyang et al., 1995*; *Schmitz et al., 1996*; *Deng et al., 2014*). For example, they prefer a large aliphatic residue immediately upstream of the target tyrosine residue in their substrates, a feature that distinguishes their specificity from that of ZAP-70, which prefers an aspartic acid residue at this position (*Shah et al., 2016*).

Given the close similarity between the Src-family kinases, we were interested in seeing whether the inability of Lck to phosphorylate LAT and SLP-76, while maintaining robust phosphorylation of ITAMs and ZAP-70, is a property that is shared with c-Src, given that c-Src is broadly-expressed and did not evolve under the constraints of T cell signaling. Also, the conclusions of our earlier paper were based on the analysis of a small set of peptides from the T cell receptor and LAT, and did not include a more general analysis of the specificity of these kinases against a broad range of potential targets.

We have now expanded our kinase specificity screening platform through construction of a genetically encoded peptide library containing ~2600 diverse sequences derived from known tyrosine phosphorylation sites in the human proteome. Using this library, we validated our earlier conclusions regarding the orthogonal specificities of Lck and ZAP-70. In comparing the specificity of Lck to that of c-Src using this library, we find a subtle, but significant, divergence in the substrate preferences of these two enzymes. While both Src-family kinases are equally efficient at phosphorylating a large group of common substrates, there are peptides that are phosphorylated efficiently by c-Src and not by Lck, and vice versa. This divergence is characterized by a suppression in the ability of Lck, relative to c-Src, to phosphorylate highly acidic phosphosites while tolerating the presence of positively charged residues in the substrates. More generally, our approach provides a tool that is

easily deployed to interrogate the proteome-wide specificity of any tyrosine kinase, by leveraging the speed and accuracy of next-generation sequencing.

## Results and discussion

### The Human-pTyr library screen, an expanded platform for determining the substrate specificity of tyrosine kinases using bacterial surface-display

We created a genetically-encoded library based on 2600 sequences, each spanning 15 residues surrounding known sites of tyrosine phosphorylation in the human proteome. Sequences were chosen by compiling a list of all of the tyrosine phosphorylation sites in human proteins annotated in the Uniprot database (*The UniProt Consortium, 2017*), along with human sequences derived from a list of experimentally validated kinase-substrate pairs from the PhosphoSitePlus database (*Hornbeck et al., 2015*). To construct the library, we purchased peptide-coding oligonucleotides generated by massively parallel on-chip DNA synthesis (Twist Bioscience), and we cloned these coding sequences into plasmids containing a bacterial surface-display scaffold gene (*Rice and Daugherty, 2008*). We refer to this library as the 'Human-pTyr' library, since we consider this library to be representative of the sequence space that is accessible to phosphorylation by the complement of human tyrosine kinases.

The Human-pTyr library comprises relatively diverse sequences (*Figure 1D* and *Figure 1—figure supplement 1*). Glutamate and aspartate residues are the most common at several positions in the peptides, consistent with the fact that generic tyrosine kinase substrates, such as poly-Glu-Tyr, are enriched in acidic residues (*Braun et al., 1984*). Nevertheless, acidic residues do not dominate any position in the library. Hydrophobic residues are under-represented at most positions, which might reflect the fact that phosphorylation sites are not usually buried in the hydrophobic cores of proteins.

In a typical screen, the surface-displayed Human-pTyr library was phosphorylated for a limited time, sufficient to achieve approximately 20–30% of the maximal possible phosphorylation. After labeling with a fluorescent pan-phosphotyrosine antibody, cells with the highest fluorescence (25% of the total cells) were isolated by cell sorting and analyzed by Illumina deep sequencing, in comparison to unsorted cells. These conditions ensure sufficient phosphorylation for a robust signal-to-noise ratio, without saturating the system and losing discrimination. Screens were typically carried out in duplicate or triplicate (*Figure 1—figure supplement 2*), and the normalized enrichment scores from independent screens were averaged before further analysis. We note that although the surface-display levels of each peptide may vary, no correction was made for this variability. In our previous study, we developed a strategy that employs cell sorting and deep sequencing to accurately measure the relative surface-display levels of all peptides in a library, taking advantage of an epitope tag attached to an extracellular region of the surface-display scaffold protein. Using this protocol, we showed previously that such a correction for surface-display level does not alter the gross features of our specificity analyses (*Shah et al., 2016*). For this study, we were primarily interested in comparisons between substrate preferences for pairs of kinases, where any variation in surface-display levels would be common to both screens.

### Screens using the Human-pTyr library confirm the expected sequence preferences of c-Abl and ZAP-70

We tested if the use of the Human-pTyr library in the bacterial surface-display system recapitulates the expected preferences for a well-studied kinase, c-Abl. The distribution of enrichment scores for phosphorylation by the c-Abl kinase domain is bimodal, with ~70% of the peptides being poorly phosphorylated and 30% of the peptides being phosphorylated relatively efficiently (*Figure 2A*). The fact that approximately a third of the peptides are phosphorylated efficiently is consistent with the view that tyrosine kinases are, in general, promiscuous enzymes, with the ability to target a broad range of potential substrates (*Miller, 2003*).

We visualized the position-dependent amino acid preferences for phosphorylation by c-Abl by generating a sequence probability logo based on sequences in the subset of peptides with high enrichment scores (*O'Shea et al., 2013*) (*Figure 2B*). We refer to this sequence logo as a 'phospho-

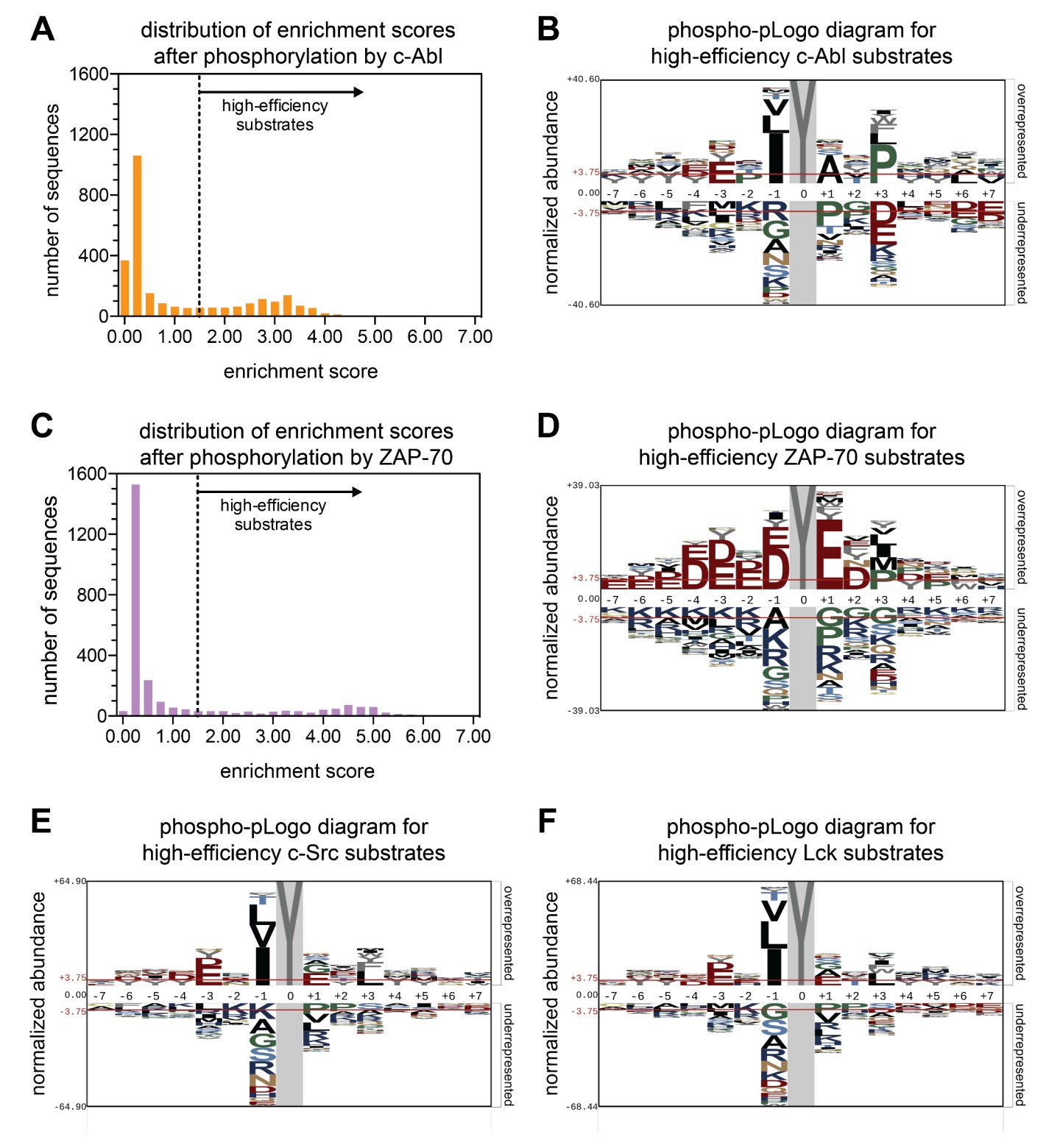

**Figure 2.** Phosphorylation of the Human-pTyr library by the kinase domains of c-Abl, ZAP-70, c-Src, and Lck. (**A**) A histogram showing the distribution of enrichment scores for 2587 peptides upon phosphorylation by c-Abl. 'High-efficiency' c-Abl substrates are defined as those with enrichment scores above the dashed line in the graph. (**B**) Phospho-pLogo diagram showing the probability at each position of the amino acids in the high-efficiency c-Abl substrates (791 peptides) relative to all substrates in the library (2587 peptides). (**C**) Histogram of enrichment score distribution, as in (**A**), for ZAP-70. (**D**) Phospho-pLogo diagram, as in (**B**), for ZAP-70 (604 high-efficiency peptides). (**E**) Phospho-pLogo diagram, as in (**B**), for c-Src (791 high-efficiency

*Figure 2 continued on next page*

*Figure 2 continued*

peptides). (F) Phospho-pLogo diagram, as in (B), for Lck (763 high-efficiency peptides). Sequence logos were visualized using pLogo (*O'Shea et al., 2013*). In each case, the same cutoff of an enrichment score greater than or equal to 1.5 was used. Data for c-Src are the average of three independent screens, and data for c-Abl, Lck, and ZAP-70 are the average of two independent screens.
DOI: https://doi.org/10.7554/eLife.35190.005

The following source data and figure supplements are available for figure 2:

**Source data 1.** Enrichment scores for all peptides analyzed in screens with c-Src (wild-type and mutants), Lck, c-Abl, and ZAP-70.
DOI: https://doi.org/10.7554/eLife.35190.008

**Figure supplement 1.** Sequences of several negatively charged tyrosine phosphorylation sites in lymphocyte scaffold proteins represented in the Human-pTyr library.
Enrichment scores for phosphorylation of these sites by ZAP-70 and Lck are show as a histogram.
DOI: https://doi.org/10.7554/eLife.35190.006

**Figure supplement 2.** Histograms showing the distributions of enrichment scores for 2587 peptides upon phosphorylation by c-Src and Lck.
'High-efficiency' substrates are defined as those with enrichment scores above the dashed line in the graph.
DOI: https://doi.org/10.7554/eLife.35190.007

pLogo diagram'. This logo reflects the probability of finding a particular amino acid at a specific position in a good substrate, relative the same residue- and position-specific probabilities in the whole Human-pTyr library. Amino acid residues with a positive phospho-pLogo score at a specific position are over-represented in the pool of efficiently phosphorylated substrates, relative to the whole library.

Previous studies have defined three strong determinants of efficient phosphorylation by the c-Abl kinase domain: a bulky aliphatic residue at the −1 position with respect to the tyrosine, an alanine at the +1 position, and a proline at the +3 position (*Songyang et al., 1995*; *Till et al., 1994*; *Till et al., 1999*; *Deng et al., 2014*). The c-Abl phospho-pLogo for the Human-pTyr library recapitulates these known sequence preferences for c-Abl (*Figure 2B*). Our analysis also provides information about sequence features that are disfavored, such as a proline residue at the +1 position in the substrate.

We also analyzed the phosphorylation of the Human-pTyr library by the ZAP-70 kinase domain. As for c-Abl, the distribution of enrichment scores is bimodal, although the separation between the inefficient and efficient substrates is more marked, with only ~20% of the peptides falling in the latter group (*Figure 2C*). As expected based on our earlier studies, the phospho-pLogo diagram for ZAP-70 reveals a strong preference for substrates enriched in aspartate and glutamate residues, with lysine or arginine residues being disfavored (*Figure 2D*). Interestingly, these sequence features are characteristic not just of phosphosites in LAT and SLP-76, but also those in other scaffold and adapter proteins in lymphocytes, such as NTAL and LAX. These proteins are known substrates of the ZAP-70 paralog Syk (*Horejsí et al., 2004*), and are robustly phosphorylated by ZAP-70 in our screen (*Figure 2—figure supplement 1*).

## A subtle divergence in the specificities of Lck and c-Src

We next analyzed the specificities of the kinase domains of two Src-family kinases, c-Src and Lck. As for c-Abl and ZAP-70, the distribution of enrichment scores is bimodal (*Figure 2—figure supplement 2*). The phospho-pLogo diagrams generated using the most efficiently phosphorylated peptides are strikingly similar for Lck and c-Src, and these logos are different from those for c-Abl and ZAP-70. The observed substrate preferences for Lck and c-Src are in line with preferences observed previously using oriented peptide libraries composed of degenerate sequences (*Figure 2E–F*) (*Songyang et al., 1995*; *Deng et al., 2014*). Both kinases had a strong preference for isoleucine, leucine, or valine at the −1 position, and a moderate preference for a bulky hydrophobic residue at the +3 position.

Based on visual inspection of the phospho-pLogo diagrams, the specificities of Lck and c-Src appear indistinguishable. A subtle distinction between Lck and c-Src emerges, however, upon comparing the efficiency of phosphorylation of individual peptides by different kinases (*Figure 3A*). Comparison of duplicate experiments with Lck shows a high degree of correlation in the efficiency of phosphorylation for each peptide in the Human-pTyr library (*Figure 3A*, first panel). On comparing Lck and c-Src, a difference is evident (*Figure 3A*, second panel). Although there is still a high degree of correlation between the two datasets, with 26 and 65% of the peptides being good or

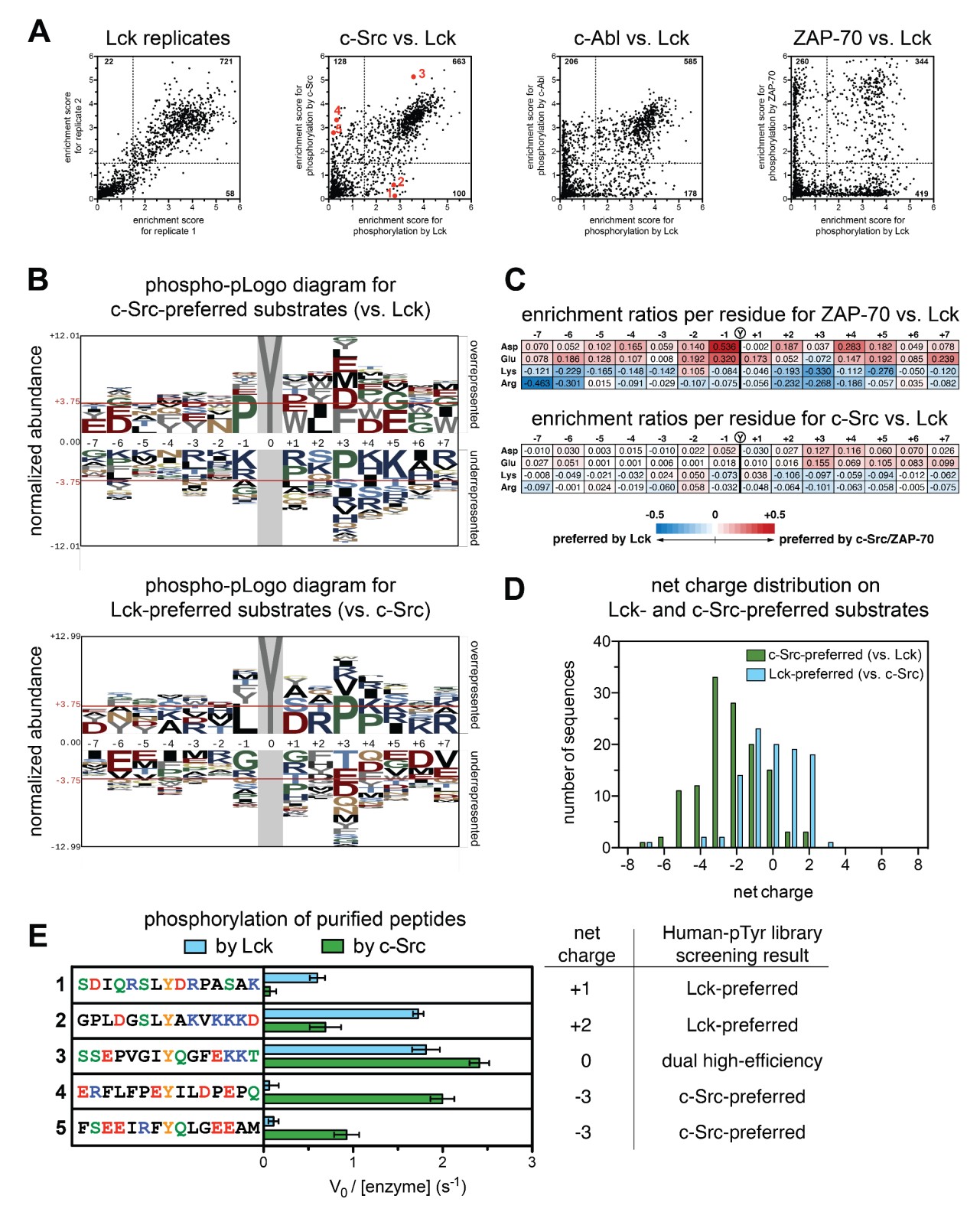

**Figure 3.** Specificity differences between closely-related kinases, c-Src and Lck. (**A**) Scatter plots showing the correlations between enrichment scores for all peptides in the Human-pTyr library from multiple screens with different kinases. *First panel:* correlation between two individual Lck replicates. *Second panel:* correlation between Lck screens (duplicate) and c-Src screens (triplicate); peptides analyzed in **Figures 3E** and **4C** are denoted by red dots. *Third panel:* correlation between Lck screens (duplicate) and c-Abl screens (duplicate). *Fourth panel:* correlation between Lck screens (duplicate)

*Figure 3 continued*

and ZAP-70 screens (duplicate). (B) Phospho-pLogo diagrams showing the probability of each amino acid per residue in the c-Src-preferred substrates (top logo), which are not efficiently phosphorylated by Lck (128 peptides), and in the Lck-preferred substrates (bottom logo), which are not efficiently phosphorylated by c-Src (100 peptides), relative to all substrates in the library. (C) Relative enrichment values for ZAP-70 or c-Src versus Lck, comparing the abundance of charged residues at each position in substrates after phosphorylation by either kinase. Enrichment values for each residue are calculated as described in Materials and methods. Relative enrichment values are expressed as $\log_{10}$ (ZAP-70 or c-Src enrichment/Lck enrichment). (D) Distribution of net charges on peptides that were selectively phosphorylated by c-Src or Lck, relative to one another, showing a greater tolerance for negatively charged substrates in c-Src relative to Lck. Net charge was estimated as the difference between the number of Lys/Arg residues and Asp/Glu residues. (E) In vitro phosphorylation kinetics of five purified peptides by the c-Src and Lck kinase domains. All peptides were used at a concentration of 250 µM, and both kinases were used at a concentration of 500 nM. The corresponding peptides analyzed in screens with c-Src and Lck are also highlighted as red dots in the second panel of *Figure 3A*. Error bars represent the standard deviation of at least three measurements.

DOI: https://doi.org/10.7554/eLife.35190.009

bad substrates, respectively, for both kinases, ~10% of the substrates were only phosphorylated efficiently by one kinase but not the other. When comparing phosphorylation by Lck to that by c-Abl and ZAP-70, the exclusion effect is much stronger, with an orthogonality in specificity most clearly demonstrated for the comparison of Lck and ZAP-70 (*Figure 3A*, third and fourth panels).

To understand the origin of the specificity difference between Lck and c-Src we generated phospho-pLogo diagrams for two groups of sequences, one containing peptides that are good Lck substrates but poor c-Src substrates (referred to as the Lck-preferred peptides), and the other having the opposite specificity (the c-Src-preferred peptides) (*Figure 3B*). For these two subsets of sequences, the phospho-pLogo diagrams show a clear alteration in specificity between the two kinases. For the c-Src-preferred peptides, there is an over-representation of negatively charged residues C-terminal to the phosphotyrosine, and an under-representation of positively charged residues (*Figure 3B*, top panel). The opposite is true for the Lck-preferred peptides (*Figure 3B*, bottom panel).

Our previous analysis of the specificity determinants of Lck phosphorylating ITAM peptides had indicated that Lck does not have strong preferences for sidechains located before the phosphotyrosine (*Shah et al., 2016*). Consistent with this finding, the features of the phospho-pLogo diagrams are less distinct for residues before the phosphotyrosine, but even so one can discern an over-representation of negatively charged residues, and an exclusion of positively charged ones in the c-Src-preferred peptides. Again, the opposite is true for the Lck-preferred peptides that are poor c-Src substrates.

Our results indicate that c-Src is more tolerant of negatively charged residues in substrates than Lck, and less tolerant of positively charged residues. This is reminiscent of the comparison between ZAP-70 and Lck, as revealed by our previous analysis. In *Figure 3C*, we show the differential enrichment scores for phosphorylation of all sequences in the Human-pTyr library by c-Src versus Lck, and by ZAP-70 versus Lck, focusing on Asp, Glu, Arg and Lys residues at each position along the peptide. The sharp differential selectivity for charged residues between ZAP-70 and Lck is also seen, in muted form, between c-Src and Lck. The peptides favored by c-Src have significantly greater net negative charge than those favored by Lck, pointing to an electrostatic basis for the differential specificity (*Figure 3D*).

We carried out in vitro measurements of phosphorylation kinetics with purified kinase domains and five representative synthetic peptides: two c-Src-preferred peptides, two Lck-preferred peptides, and one peptide that is a good substrate for both kinases in the screening data (*Figure 3A*, second panel). Consistent with our screens, the c-Src-preferred peptides were readily phosphorylated by c-Src but not by Lck, both Lck-preferred peptides were phosphorylated efficiently by Lck, but with either reduced or negligible efficiency by c-Src, and the predicted common substrate was phosphorylated robustly by both kinases (*Figure 3E*).

## Changes in specificity-determining residues in the two branches of the Src-family kinases

Lck has presumably evolved to suppress off-target phosphorylation of the highly negatively charged ZAP-70 substrates in T cells, LAT and SLP-76. Given that similar substrates exist in other hematopoietic cell types, where signaling also requires an interplay between Src- and Syk-family kinases (*Weiss and Littman, 1994*; *Lowell, 2011*), this electrostatic fine-tuning may be a conserved feature

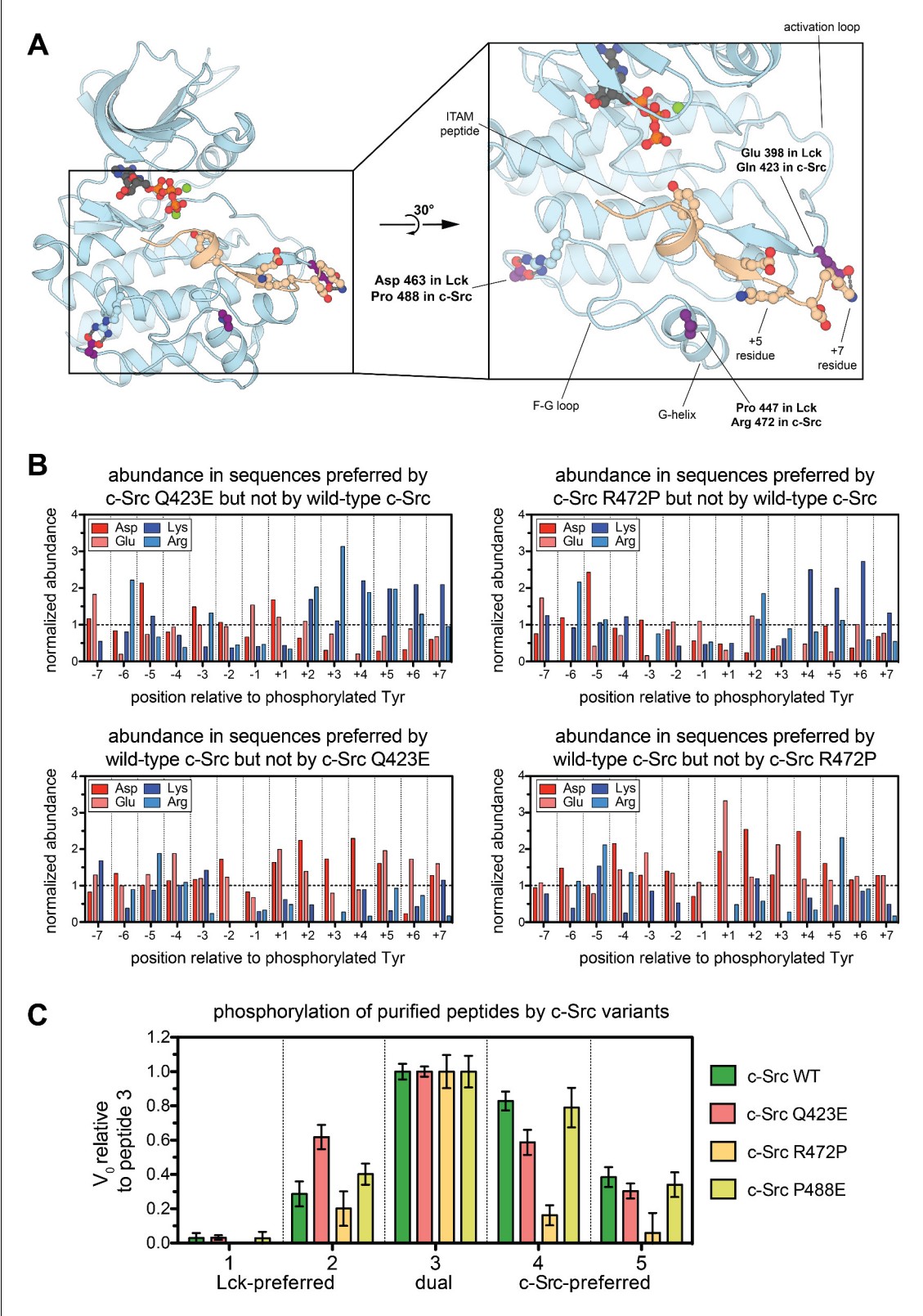

**Figure 4.** Conserved electrostatic differences between Src-A and Src-B kinases. (A) A representative instantaneous structure from a simulation of the Lck kinase domain bound to an ITAM peptide (TCRζ residues 104–118), highlighting the position of the three residues analyzed in *Figure 4—figure supplement 1*. (B) Histograms showing the prevalence of charged residues at each position in the subset of peptides that are preferred substrates of c-Src Q423E or R472P, but not wild-type c-Src (top) and wild-type c-Src but not c-Src Q423E or R472P (bottom). The abundances of these charged

*Figure 4 continued on next page*

*Figure 4 continued*

residues are normalized to those found in the whole Human-pTyr library. (**C**) Comparison of the in vitro rates for phosphorylation of the five peptide sequences in *Figure 3E* by four c-Src kinase domain variants: wild-type c-Src and the Q423E, R472P, and P488E mutants. All peptides were used at a concentration of 250 µM, and all kinases were used at a concentration of 500 nM. For each kinase, all rates were normalized to the rate of phosphorylation of peptide 3, and error bars represent the standard deviation from at least three measurements. The absolute rate constants are plotted in *Figure 4—figure supplement 4*.

DOI: https://doi.org/10.7554/eLife.35190.010

The following figure supplements are available for figure 4:

**Figure supplement 1.** Sequence logos for the regions surrounding three residues that may cause divergent substrate specificities between Src-A and Src-B kinases.

DOI: https://doi.org/10.7554/eLife.35190.011

**Figure supplement 2.** Correlations between enrichment scores for all sequences in the Human-pTyr library upon phosphorylation between wild-type c-Src and the c-Src mutants Q423E, R472P, and P488E.

DOI: https://doi.org/10.7554/eLife.35190.012

**Figure supplement 3.** Phospho-pLogo diagrams for preferred substrates of c-Src Q423E, c-Src R472P, and c-Src P488E that are not efficiently phosphorylated by wild-type c-Src.

DOI: https://doi.org/10.7554/eLife.35190.013

**Figure supplement 4.** Initial velocities for phosphorylation of the five peptide sequences shown in *Figure 3E* by wild-type c-Src and three c-Src mutants, Q423E, R472P, and P488E.

DOI: https://doi.org/10.7554/eLife.35190.014

**Figure supplement 5.** Initial velocities for phosphorylation of the five peptide sequences shown in *Figure 3E* by two Src-A kinase domains, c-Src and Fyn, and two Src-B kinase domains, Lck and Hck.

DOI: https://doi.org/10.7554/eLife.35190.015

that differentiates the Src-A and Src-B lineages. We analyzed the sequences of Src-family kinase domains across vertebrates and identified three positions near the substrate docking site where the charge on the residue is maintained within the Src-A and Src-B families, but not between the two groups (*Figure 4A* and *Figure 4—figure supplement 1*).

One of these residues, Glu 398 in Lck (Gln 423 in c-Src) lies in the activation loop and is poised to sense charged residues downstream of the substrate tyrosine. In a 500 ns molecular dynamics simulation of Lck bound to a preferred substrate derived from an ITAM motif, Glu 398 repeatedly forms ion pairs with an arginine residue downstream of the ITAM tyrosine (*Figure 4A*). Negatively charged substrate residues in this vicinity would presumably weaken binding to the Lck activation loop, but not to that of c-Src, and positively charged residues should be preferred by Lck. In c-Src, a proximal residue on the G-helix, Arg 472 (Pro 447 in Lck), may assist in the coordination of substrates with negative charge located after the tyrosine (*Figure 4A*). The third residue of interest, Asp 463 in Lck (Pro 488 in c-Src), is unlikely to make direct contact with substrates. In simulations of Lck, Asp 463 neutralizes an arginine residue on the F-G loop, which is unpaired in crystal structures of c-Src (*Figure 4A*). The F-G loop of tyrosine kinases can control electrostatic selectivity upstream of tyrosine residues in substrates (*Shah et al., 2016*), and this unpaired arginine in Src-A kinases may explain the slight preference exhibited by c-Src, relative to Lck, for negative charge upstream of the tyrosine.

We tested the impact of three mutations in c-Src, Q423E, R472P, and P488E, on the phosphorylation of the Human-pTyr library. Enrichment scores from these screens correlated strongly with those from the wild-type c-Src screen, with few substrates being selectively phosphorylated by either wild-type c-Src or a mutant (*Figure 4—figure supplement 2*). Each mutant efficiently phosphorylated ~50 peptides that were not efficiently phosphorylated by wild-type c-Src. Analysis of peptides that were selectively phosphorylated by the c-Src Q423E and R472P mutants, but not wild-type c-Src, showed a clear enrichment in sequences with positive charge downstream of the tyrosine (*Figure 4B* and *Figure 4—figure supplement 3*). The c-Src P488E mutant showed a similar, albeit less clear, preference (*Figure 4—figure supplement 3*).

We measured phosphorylation kinetics for these mutants using the same panel of five purified peptides described above, which have varying degrees of charge (*Figure 3E*). To account for differences in the catalytic activity of each mutant, phosphorylation rates are depicted relative to the values for the dual c-Src/Lck substrate, peptide 3, in *Figure 4C*, and absolute rates are given in

*Figure 4—figure supplement 4*. The Q423E mutant c-Src gained substantial activity relative to wild-type c-Src toward an Lck-preferred substrate (peptide 2) that contains several positively charged residues downstream of the tyrosine, consistent with our structural model. The R472P mutant c-Src mutant had reduced activity against all five peptides tested (*Figure 4—figure supplement 4*); however, this mutant preferentially lost activity against the negatively-charged c-Src-preferred peptides (*Figure 4C*). The P488E c-Src mutant displayed activity similar to wild-type c-Src against these five substrates, suggesting that this residue may not play a strong role in differentiating the specificities of c-Src and Lck.

We also compared the specificities of c-Src and Lck to that of Fyn and Hck, which are Src-A and Src-B kinases, respectively (*Figure 1C*). Fyn is expressed in many cell types, like c-Src, and it is the second most prominent Src-family kinase in T cells, after Lck. We found that Fyn had intermediate specificity between c-Src and Lck when analyzed with the panel of five purified peptides described above (*Figure 4—figure supplement 5*). The Src-B kinase Hck showed the expected Lck-like specificity using the same peptides. The unexpected divergence between the Src-A kinases c-Src and Fyn may reflect yet unknown selective pressures on the substrate specificity of Fyn.

## Additional insights from specificity screens using the Human-pTyr library

In this report, we focused primarily on the subtle electrostatic differences between the Src-family kinases Lck and c-Src, and the implications of these differences for our understanding of T cell receptor signaling. The high-throughput screens using the Human-pTyr Library also provide additional insights into other aspects of tyrosine kinase specificity. For example, in comparing c-Src and Lck, it is evident that these kinases not only differ in their preference for charged residues, but they also have slightly different preferences at the −1 and +3 positions on substrates (*Figures 3B* and *5A*). Tyrosine kinases are generally most selective with respect to these substrate positions, and many kinases have distinct preferences at these positions (*Figure 5A* and [*Songyang et al., 1995*]).

The specificity screens and molecular dynamics simulations of Lck and ZAP-70 presented here and in our previous report (*Shah et al., 2016*) provide insights into the structural determinants of −1 selectivity. A residue on the F-G loop that lines the catalytic cleft makes direct contacts with −1 residues in both Lck and ZAP-70 simulations (*Figure 5B* and [*Shah et al., 2016*]). This F-G loop residue is an isoleucine in Lck and valine in c-Src, consistent with their strong but slightly different preferences for hydrophobic −1 residues (*Figure 3B*). c-Abl has a serine residue at this position, which may explain its slightly broadened −1 residue tolerance, and ZAP-70 has a lysine residue, consistent with a strong preference of this kinase for a −1 aspartate or glutamate residue (*Figure 5A*). Mutation of this F-G loop residue in c-Src alters −1 residue preferences in accordance with these observations (*Figure 5—figure supplement 1*). Our simulations, and previous reports (*Till et al., 1999*), have also identified specific residues in the +3 pocket that differentiate the four kinases described here (*Figure 5B*); however, the precise rules defining +3 residue preferences are not clear.

The high-throughput screens presented here were carried out with a library of diverse, discrete peptide sequences derived from the human proteome. As such, they provide an opportunity analyze kinase specificity in ways that are not accessible to degenerate library approaches. For example, analysis of discrete sequences allows for the identification of independent linear motifs that are buried in the phospho-pLogo diagrams, and to assess the interdependence of specific residue preferences (*Kettenbach et al., 2012*). We analyzed the preferred c-Src, Lck, c-Abl, and ZAP-70 substrates using the tool Motif-X to extract the independent linear motifs that collectively comprise the phospho-pLogo diagrams (*Schwartz and Gygi, 2005*; *Chou and Schwartz, 2011*). This analysis reproduced the known specificity determinants for all four kinases, primarily at the −3, −1, +1, and +3 positions (*Figure 5—source data 1*). Interestingly, all of the statistically significant independent motifs for these four kinases were defined by single position preferences, suggesting negligible interdependence between favorable sequence features at these positions. It is noteworthy, however, that the analysis using Motif-X cannot describe conditional negative selection. It is possible that the presence of a specific amino acid residue at one site may make certain residues unfavorable at another site.

Screening results using the Human-pTyr library can also be directly compared to data that describe interactions between signaling partners in a biological context. We compared the enrichment scores for specific kinase-substrate pairs in the bacterial surface-display screens with a curated

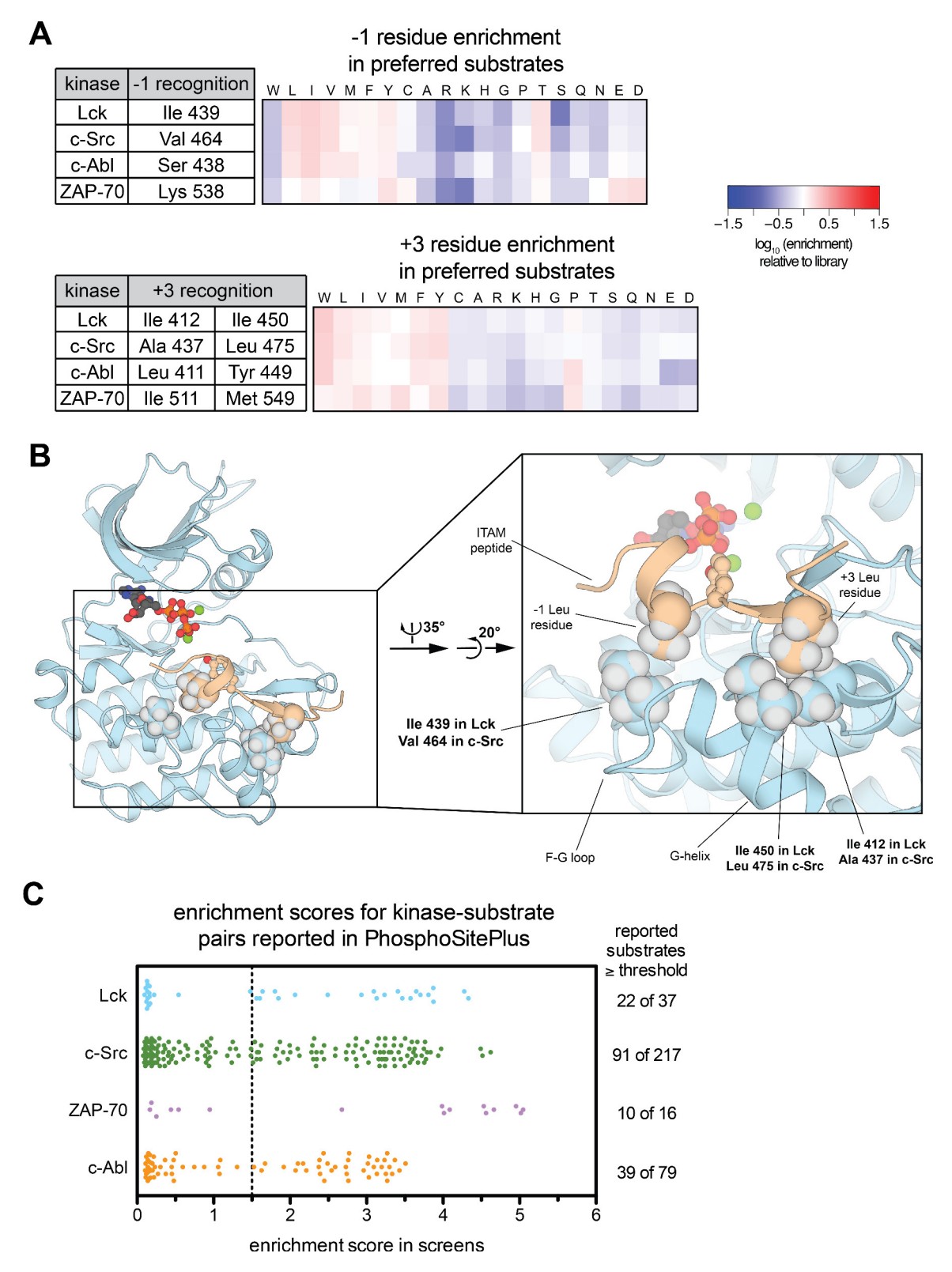

**Figure 5.** Additional insights from high-throughput specificity screens. (**A**) Heatmaps depicting the enrichment of −1 and +3 residues in the preferred substrates of Lck, c-Src, c-Abl, and ZAP-70. Enrichment is calculated relative to the amino acid abundance at these positions in the whole Human-pTyr library, analogous to the depictions in *Figure 2*. Residues in each kinase that may confer specificity at these positions are given in the tables adjacent to the heatmaps. (**B**) A representative instantaneous structure from a simulation of the Lck kinase domain bound to an ITAM peptide (TCRζ residues

*Figure 5 continued on next page*

*Figure 5 continued*

104–118), highlighting the position of the three residues in the kinase domain listed in panel A that make contacts with the −1 and +3 positions on substrates. (C) Comparison of results from specificity screens with the Human-pTyr Library to a curated list of human kinase-substrate pairs from the PhosphoSitePlus database (*Hornbeck et al., 2015*). The scatter plot shows the enrichment scores for tyrosine phosphosites that were assigned to each specific kinase and also present in the Human-pTyr Library. The number of sequences greater than or equal to a threshold enrichment score of 1.5 is listed to the right of the scatter plot.

DOI: https://doi.org/10.7554/eLife.35190.016

The following source data and figure supplement are available for figure 5:

**Source data 1.** Independent linear phosphorylation motifs extracted from screens of the Human-pTyr library using Motif-X (*Schwartz and Gygi, 2005*; *Chou and Schwartz, 2011*).
DOI: https://doi.org/10.7554/eLife.35190.018

**Source data 2.** Comparison of bacterial surface-display screens with reported kinase-substrate pairs described in a curated list from the PhosphoSite-Plus database (*Hornbeck et al., 2015*).
DOI: https://doi.org/10.7554/eLife.35190.019

**Figure supplement 1.** Initial velocities for phosphorylation of LAT-based peptides by c-Src variants with F-G loop mutations.
DOI: https://doi.org/10.7554/eLife.35190.017

set of kinase-substrate pairs from the PhosphoSitePlus database, where many of the interactions were reported to have been observed in an in vivo setting (*Hornbeck et al., 2015*). In total, for c-Src, Lck, c-Abl, and ZAP-70, 349 kinase-substrate pairs were listed in the database and also measured in the screens (*Figure 5—source data 2*). For each kinase, only 40–60% of the reported phosphosites in the curated PhosphoSitePlus list were efficiently phosphorylated in the bacterial surface-display setting (*Figure 5C*). This disparity raises the question of what other factors may play a role in dictating kinase specificity in vivo.

The bacterial surface-display screens faithfully recapitulate phosphorylation efficiencies measured with purified peptides and kinases (*Shah et al., 2016*). The inability of these screens to fully recapitulate reported biologically relevant kinase-substrate interactions can be explained in at least three ways. (1) Few databases of kinase-substrate pairs are completely accurate, as it remains challenging to unambiguously determine in vivo substrates of protein kinases. (2) Substrate selection in vivo is not only driven by primary sequence recognition in the kinase active site, but also by interactions outside of the active site that dictate sub-cellular localization. (3) Natural substrates of kinases are not necessarily optimized for maximal phosphorylation efficiency. Despite the observed discrepancies between our screens and a database of biologically-relevant interactions, the data presented here, and in our previous manuscript (*Shah et al., 2016*), support the notion that sequence recognition by tyrosine kinase domains can play an important role in shaping signaling pathways.

## Concluding remarks

There is a strong correlation between the substrate preferences and evolutionary relationships of kinases: closely related kinases typically have similar sequence preferences (*Songyang et al., 1995*; *Deng et al., 2014*; *Schmitz et al., 1996*). Differences in specificity between the Src-family kinases have been observed previously, using surface-plasmon resonance detection of phosphorylation in a broad range of potential substrates, although a clear picture of the structural differences underlying the observed specificity differences did not emerge from that study (*Takeda et al., 2010*). We now show that there is a subtle divergence in substrate specificities within the Src-family kinases, whereby Lck disfavors the phosphorylation of negatively charged substrates relative to c-Src. Our analyses suggest that this electrostatic preference reflects a branch-point in the evolution of Src-family kinases, with the Src-B lineage specialized to act in immune cells in coordination with Syk-family kinases, and the Src-A lineage acting broadly across a variety of cell types. More generally, our results demonstrate how the substrate-recognition properties of closely related kinase domains can be altered subtly over the course of evolution to refine specificity in response to specific functional requirements.

Our discovery of fine-tuning within Src-family kinases stems directly from the use of a high-throughput specificity screen and a peptide library encoding thousands of sequences derived from human tyrosine phosphorylation sites. The high-throughput method to interrogate this library can be applied to the analysis of the specificity of any tyrosine kinase and is easy to deploy. The power of the method derives from the increased speed and accuracy of modern sequencing methods. We

envision that our approach will not only guide our understanding of sequence recognition by tyrosine kinase domains, as described in this study, but also aid in the identification of unknown tyrosine kinase-substrate interactions in phosphotyrosine signaling pathways.

# Materials and methods

| Reagent type (species) or resource | Designation | Source or reference | Identifiers | Additional information |
|---|---|---|---|---|
| Strain, strain background (E. coli) | TOP10 | Thermo Fisher Scientific | Thermo: C404006 | bacterial cells used for library cloning |
| Strain, strain background (E. coli) | MC1061 | Coli Genetic Stock Center | CGSC: 6649 | bacterial cells used for surface-display screens |
| Strain, strain background (E. coli) | BL21(DE3) | Thermo Fisher Scientific | Thermo: C600003 | bacterial cells used for protein over-expression |
| Strain, strain background (E. coli) | DH10Bac | Thermo Fisher Scientific | Thermo: 10361012 | bacterial cells used to produce bacmids for baculovirus production |
| Cell line (Spodoptera frugiperda) | Sf21 | UC Berkeley Cell Culture Facility | | insect cells used for baculovirus-mediated protein over-expressoin |
| Antibody | anti-phosphotyrosine-PE, recombinant clone 4G10 | Millipore | Millipore: FCMAB323PE | phycoerythrin-labeled mouse monoclonal pan-phosphotyrosine antibody (1:50 dilution) |
| Recombinant DNA reagent | pBAD33-eCPX | PMID:18480093 | Addgene: 23336 | plasmid vector for bacterial surface-display screens encoding the eCPX scaffold |
| Recombinant DNA reagent | pBAD33-eCPX-cStrep | PMID:27700984 | | plasmid vector for bacterial surface-display screens encoding the eCPX scaffold modified wth a C-terminal Strep-tag |
| Recombinant DNA reagent | Human pTyr Library | this report | | Library of pBAD33 plasmids containing ~ 2600 peptide-coding sequences fused to the 5' end of the eCPX scaffold for bacterial surface-display |
| Recombinant DNA reagent | pFastBac-ZAP-70 (KD)-His6 | PMID:27700984 | | bacmid-generating vector encoding the human ZAP-70 kinase domain (residues 327–606) with a C-terminal His6-tag |

*Continued on next page*

Continued

| Reagent type (species) or resource | Designation | Source or reference | Identifiers | Additional information |
|---|---|---|---|---|
| Recombinant DNA reagent | pFastBac-His6-TEV-Lck(KD) | PMID:27700984 | | bacmid-generating vector encoding the human Lck kinase domain (residues 229–509) with an N-terminal His6-tag and TEV protease recognition sequence |
| Recombinant DNA reagent | pET-23a-His6-TEV-Abl(KD) | this report | | bacterial expression vector encoding the mouse c-Abl kinase domain (residues 232–502) with an N-terminal His6-tag and TEV protease recognition sequence |
| Recombinant DNA reagent | pET-23a-His6-TEV-Src(KD) | this report | | bacterial expression vector encoding the chicken c-Src kinase domain (residues 257–525), wild-type sequence or mutants described in the main text, with an N-terminal His6-tag and TEV protease recognition sequence |
| Recombinant DNA reagent | pET-23a-His6-TEV-Fyn(KD) | this report | | bacterial expression vector encoding the human Fyn kinase domain (residues 261–529) with an N-terminal His6-tag and TEV protease recognition sequence |
| Recombinant DNA reagent | pET-23a-His6-TEV-Hck(KD) | this report | | bacterial expression vector encoding the human Hck kinase domain (residues 252–520) with an N-terminal His6-tag and TEV protease recognition sequence |
| Sequence-based reagent | Human pTyr Oligo Pool | Twist Bioscience | | pool of ~ 3500 80 mer DNA oligonucleotides used to generate the Human pTyr Library |
| Sequence-based reagent | eCPX-Library-5'-Fwd | Integrated DNA Technologies | | forward DNA primer used to amplify the Human pTyr Oligo Pool, sequence: 5'-ACTTCCGTAGCTGG CCAGTCTGGCCAG-3' |
| Sequence-based reagent | eCPX-Library-3'-Rev | Integrated DNA Technologies | | reverse DNA primer used to amplify the Human pTyr Oligo Pool, sequence: 5'-CAGACTGCCCA GACTGCCCTCC-3' |

*Continued*

| Reagent type (species) or resource | Designation | Source or reference | Identifiers | Additional information |
|---|---|---|---|---|
| Sequence-based reagent | TruSeq-eCPX-Fwd | Integrated DNA Technologies | | forward DNA primer used for amplification of peptide-coding DNA after cell sorting, sequence: 5'-TGACTGGAGTTCAG ACGTGTGCTCTTCCG ATCTNNNNNNNACCGC AGGTACTTCCGTAGCT-3' |
| Sequence-based reagent | TruSeq-eCPX-Rev | Integrated DNA Technologies | | reverse DNA primer used for amplification of peptide-coding DNA after cell sorting, sequence: 5'- CACTCTTTCCCTACA CGACGCTCTTCCGA TCTNNNNNNTTTT GTTGTAGTCACCAGACTG-3' |
| Peptide, recombinant protein | ZAP-70 | PMID:27700984 | | human ZAP-70 kinase domain (residues 327–606) with a C-terminal His6-tag |
| Peptide, recombinant protein | Lck | this report | | human Lck kinase domain (residues 229–509) |
| Peptide, recombinant protein | c-Abl | this report | | mouse c-Abl kinase domain (residues 232–502) |
| Peptide, recombinant protein | c-Src | this report | | chicken c-Src kinase domain (residues 257–525) wild-type sequence and mutants described in the main text |
| Peptide, recombinant protein | Fyn | this report | | human Fyn kinase domain (residues 261–529) |
| Peptide, recombinant protein | Hck | this report | | human Hck kinase domain (residues 252–520) |
| Peptide, recombinant protein | Peptide 1 | Elim Biopharmaceuticals | | sequence: Ac-SDIQRSLYDRP ASAK-NH2 |
| Peptide, recombinant protein | Peptide 2 | Elim Biopharmaceuticals | | sequence: Ac-GPLDGSLYAKVK KKD-NH2 |
| Peptide, recombinant protein | Peptide 3 | Elim Biopharmaceuticals | | sequence: Ac-SSEPVGIYQGFE KKT-NH2 |
| Peptide, recombinant protein | Peptide 4 | Elim Biopharmaceuticals | | sequence: Ac-ERFLFPEYILDP EPQ-NH2 |
| Peptide, recombinant protein | Peptide 5 | Elim Biopharmaceuticals | | sequence: Ac-FSEEIRFYQLG EEAM-NH2 |
| Commercial assay or kit | Quant-iT dsDNA Assay Kit | Thermo Fisher Scientific | Thermo: Q33130 | Picogreen DNA quantification reagents for MiSeq library preparation |
| Commercial assay or kit | MiSeq Reagent Kit v2 (300-cycles) | Illumina | Illumina: MS-102-2002 | Illumina reagents for deep sequencing |

## Preparation of Human-pTyr surface-display library for tyrosine kinase specificity profiling

The sequences in the Human-pTyr library were derived from two sources. First, we compiled a list of all phosphotyrosine sites in human proteins annotated in the Uniprot database (*The UniProt Consortium, 2017*). Second, we obtained a curated list of experimentally analyzed

kinase-substrate pairs from the PhosphoSitePlus database (*Hornbeck et al., 2015*) and filtered this list to only include human tyrosine phosphorylation sites. The two lists were combined, and redundant sequences were removed, to yield ~2600 sequences of 15 residues spanning each unique phosphosite. For those phosphosites within seven residues from the N- or C-termini of their source proteins, the peptide sequence in the library was padded with glycine residues to maintain a uniform length of 15 residues per peptide, with the target tyrosine at the central position.

In the final list, ~900 sequences had more than one tyrosine residue, and so a second version of those sequences was also included in which all tyrosines except for the central tyrosine were substituted with alanine. Finally, 12 control sequences were generated with a random composition of amino acids. Three versions of each of these sequences, with a central tyrosine, phenylalanine, and alanine, were included in the peptide list. In total, our designed library contains approximately 3500 15-residue peptide sequences, which were converted into peptide-coding DNA sequences using the most prevalent codon for each amino acid in *E. coli*. The sequences were not further modified to optimize GC-content or melting temperatures. The 5' and 3' flanking sequences 'GCTGGCCAGTC TGGCCAG' and 'GGAGGGCAGTCTGGGCAGTCTG', respectively, were appended onto each peptide-coding DNA sequence. All these sequences were purchased as an oligonucleotide pool generated by massively parallel synthesis (Twist Bioscience, San Francisco, CA).

To ensure low PCR amplification bias across this diverse library, the synthesized oligonucleotide pool was amplified with the AccuPrime *Taq* DNA polymerase using a slow ramping speed (2°C/sec), long denaturing times, and only 15 cycles, as described for other diverse libraries (*Aird et al., 2011*). The DNA was amplified using the eCPX-Library-5'-Fwd primer (5'-ACTTCCGTAGCTGGCCAGTCTGGCCAG-3') and the eCPX-Library-3'-Rev primer (5'-CAGACTGCCCAGACTGCCCTCC-3'). In parallel, the gene sequence for the eCPX surface-display scaffold protein with a C-terminal Strep-tag was amplified by PCR from the pBAD33-eCPX-cStrep plasmid (*Rice and Daugherty, 2008*; *Shah et al., 2016*). The two PCR products were fused in a third PCR reaction using the AccuPrime *Taq* DNA polymerase to append the peptide-coding sequences in-frame with the 5' end of the eCPX scaffold. This library-scaffold insert was digested with the SfiI restriction endonuclease for 4 hr at 50°C and then purified over a spin column to remove the short, digested ends. The digested library-scaffold insert was ligated into an SfiI-digested pBAD33-eCPX plasmid using T4 DNA ligase overnight at 18°C. After ligation, the DNA was purified and concentrated over a spin column, then used to transform TOP10 cells by electroporation. The transformed TOP10 cells were grown in liquid culture overnight, and the plasmid library DNA was extracted and purified by miniprep the following day.

The composition of the final purified Human-pTyr library was analyzed by Illumina deep sequencing. In this library, 1911 wild-type human phosphosites, 676 mutant sequences in which non-central tyrosines were substituted with alanine, and the random control sequences described above, had sufficient abundance for accurate analysis by deep sequencing. The remaining ~25% of the sequences encoded in our purchased oligonucleotide pool had low or no read counts, suggesting that they were either not abundant in the synthesized pool or were not efficiently amplified by PCR. These low abundance sequences were not included in the analysis of any experiments described in this report. Importantly, the ~2600 peptide-coding sequences that were sufficiently-abundant in the library accounted for 82% of the deep sequencing reads, and the majority of the remaining 18% of reads could be attributed to single nucleotide insertions or deletions, most likely due to errors in synthesis. These sequences containing indels were also ignored in all analysis.

In this report, the control sequences were only used for qualitative assessment of the library selection experiments and played no role in any quantitative analysis. For this study, both the multi-tyrosine and mutant single-tyrosine variants of sequences were included in all analyses. Although not pertinent here, separate analysis of these sequences may be useful in cases where tyrosine residues outside of the target phosphorylation site play a role in sequence recognition by a tyrosine kinase, or in cases where the roles of individual tyrosine residues in specific sequences with multiple tyrosines need to be dissected.

## Experimental procedure for high-throughput specificity screens

High-throughput specificity screens using the Human-pTyr surface-displayed peptide library were carried out essentially as described in our previous study (*Shah et al., 2016*), with the primary differences being in data analysis and library preparation for deep sequencing. Briefly, *E. coli* MC1061 cells were transformed with the library plasmids and grown in liquid media to an optical density of

0.5 at 600 nm. Expression of the surface-displayed peptides was induced with 0.4% arabinose for 4 hr at 25°C. After expression, cells were harvested and washed with phosphate-buffered saline. All phosphorylation reactions were done at 37°C with 500 nM kinase domain and 1 mM ATP in a buffer containing 50 mM Tris, pH 7.5, 150 mM NaCl, 5 mM $MgCl_2$, 1 mM TCEP, and 2 mM sodium orthovanadate. The cell density in all phosphorylation reactions was kept consistent, with an optical density of approximately 1.5 at 600 nm. To determine optimal phosphorylation times, phosphorylation levels were monitored at a variety of time points. Reaction times for the screens were chosen to achieve approximately 20–30% of the maximal possible phosphorylation of the library: 3 min for all c-Src variants and Lck, and 30 min for ZAP-70 and c-Abl. Reactions were quenched with 25 mM EDTA, then the cells were washed with phosphate-buffered saline containing 0.2% BSA. Phosphorylated cells were labeled with a phycoerythrin-conjugated pan-phosphotyrosine antibody, 4G10-PE (Millipore), at a 1:50 dilution, then washed again, and subject to fluorescence-activated cell sorting.

For each screen, ~4,000,000 cells from the top 25% of the phycoerythrin fluorescence distribution were collected into 50 mL conical tubes containing 10 mL of LB medium. An aliquot of ~4,000,000 unsorted cells was prepared identically, as a reference sample. The cells were centrifuged at 5000 *g* for 30 min, and the supernatant was aspirated off, leaving behind approximately 500 µL of supernatant. We note that the cell pellets were not visible. The cells were resuspended in the remaining supernatant, transferred to 1.5 mL microcentrifuge tubes, and centrifuged at approximately 20,000 rcf for 10 min. The remaining supernatant was carefully aspirated off. The cells were resuspended in 100 µL of water, lysed by boiling for 10 min, then centrifuged again for 10 min at 20,000 rcf. 20 µL of the supernatant from this lysate was used in a 50 µL 15-cycle PCR reaction with the AccuPrime *Taq* DNA polymerase and the following forward and reverse primers: TruSeq-eCPX-Fwd (5'-TGAC TGGAGTTCAGACGTGtgctcttccgatctNNNNNNaccgcaggtacttccgtagct-3') and TruSeq-eCPX-Rev (5'-CACTCTTTCCCTACACGACgctcttccgatctNNNNNNttttgttgtagtcaccagactg-3'), where 'N' refers to a degenerate base at which a mixture of A, T, G, and C are included. 1–5 µL of the reaction solution from this first PCR reaction was used as the template for a second 100 µL 15-cycle PCR reaction with the AccuPrime *Taq* DNA polymerase and Illumina D700 and D500 series adapters. At this stage, distinct eight base pair indices were included for multiplexed sequencing on one flow cell. The resulting PCR products were purified by gel extraction. Concentrations of all samples were determined using the PicoGreen reagent (Thermo Fisher Scientific, Waltham, MA), pooled to achieve equal molarity for each sample, and sequenced by paired-end Illumina sequencing. In all cases, samples were loaded on the sequencing flow cell to obtain 500–1000 reads per peptide-coding sequence in the DNA sample derived from unsorted cells.

## Analysis of deep sequencing data from high-throughput specificity screens

The raw deep sequencing data were processed as described in our previous report (*Shah et al., 2016*), by first assembling the paired-end reads into a single contiguous read, then removing all adapter sequences to leave behind only the peptide-coding region. The trimmed DNA sequences were translated into peptide sequences using a Python script (source code provided). Then, the abundance of each peptide found in the Human-pTyr library (listed unannotated in a file called 'peptide_list.txt') was determined using the following shell command:

```
while read string ; do grep -c "$string" sample.translate.fastq ;
done < peptide_list.txt > peptide_count
```

In every sample, the raw read-counts for each peptide ($n_{peptide}$) were converted into frequencies ($f_{peptide}$) by dividing by the total number of reads in that sample that mapped to a sequence in the full library ($n_{total}$). These frequencies were converted into 'enrichment scores' by dividing the frequency of a variant in a sorted sample by the frequency in an unsorted sample prepared on the same day. The enrichment scores from replicate screens with the same kinase were averaged prior to any further analysis.

$$f_{peptide} = \frac{n_{peptide}}{n_{total}}$$

$$enrichment\ score = \frac{f_{peptide,sorted}}{f_{peptide,\ unsorted}}$$

To produce the phospho-pLogo diagrams, we first determined the distribution of enrichment scores over the whole Human-pTyr library for each kinase. These distributions are parsed with a bin size of 0.25 and are plotted in *Figure 2*. Based on the bimodal nature of the enrichment score distributions, the high-efficiency substrates for all kinases were selected as those peptides with an enrichment score ≥1.5. This subset of the Human-pTyr library, which is not weighted in any way based on the enrichment scores, was used as the 'foreground' for the generation of phospho-pLogo diagrams, and the full list of sequences in the Human-pTyr library was used as the 'background' (*O'Shea et al., 2013*). For the phospho-pLogo diagrams depicting those peptides that were preferred by Lck or c-Src but not the other (*Figure 3B*), and for those peptides preferred by c-Src mutants but not wild-type c-Src (*Figure 4—figure supplement 3*), we used as the 'foreground' dataset those sequences which had an enrichment score ≥1.5 for one kinase but <1.5 for the other kinase. The 'background' dataset in these diagrams was still the full list of sequences in the Human-pTyr library.

The enrichment ratios in *Figure 3C* are weighted values derived from the raw data for all sequences in the Human-pTyr library. To calculate these ratios, we first determined the frequency of sequences ($f_{i,j}$) in a sample that contained a residue, $i$, at a specific position, $j$. The frequency in the sorted sample was normalized by dividing the frequency in the unsorted sample to give a residue-specific enrichment score ($E_{i,j}$).

$$E_{i,j} = \frac{f_{i,j,sorted}}{f_{i,j,\ unsorted}}$$

These values were calculated for independent replicates and then averaged. The enrichment ratios given in *Figure 3C* correspond to the logarithm of the ratios of c-Src or ZAP-70 residue-specific enrichment scores over the Lck residue-specific enrichment score.

$$enrichment\ ratio = \log_{10}\left(\frac{E_{i,j,Src/ZAP}}{E_{i,j,Lck}}\right)$$

*Figure 4B* shows the abundance of specific amino acid residues at each position in the subset of sequences that are efficiently phosphorylated by c-Src Q423E or R472P but not wild-type c-Src. The values in these histograms, like the phospho-pLogo diagrams, are not weighted based on enrichment scores. To calculate these abundances, we determined the percentage of sequences in the mutant-only subset that contain the charged residue of interest at specific position, and we divided this by the percentage of sequences with the same property in the full Human-pTyr library. For example, approximately 8% of the sequences in the full Human-pTyr library have a lysine residue at the +5 position, and 16% of the sequences that are efficiently phosphorylated by c-Src Q423E but not wild-type c-Src have this same feature. Thus, the normalized abundance for a + 5 position lysine residue in this dataset is 2, as seen in *Figure 4B*.

## Purification of tyrosine kinase domains

The human ZAP-70 tyrosine kinase domain, residues 327–606, with an uncleavable C-terminal His$_6$-tag, was expressed in Sf21 insect cells and purified as described previously (*Shah et al., 2016*). Briefly, after over-expression, the cells were lysed using a cell homogenizer, then the lysate was treated with DNase1 and then clarified by ultracentrifugation. The protein was purified in three steps, first by nickel affinity chromatography, then cation exchange, and finally size exclusion chromatography over a Superdex 200 column. For all proteins, the final purification step was carried out in a buffer containing 10 mM HEPES, pH 7.5, 150 mM NaCl, 1 mM TCEP, 5 mM MgCl$_2$, and 10% glycerol. Pure proteins were concentrated to approximate ~50 µM, flash-frozen in liquid nitrogen, and stored at −80°C.

The human Lck kinase domain construct, spanning residues 229–509, with an N-terminal His$_6$-tag followed by a TEV protease cleavage site, was also expressed in Sf21 insect cells. After cell lysis using a cell homogenizer, the lysate was clarified by ultracentrifugation. YopH tyrosine phosphatase was added to the supernatant to a concentration of ~20 nM to dephosphorylate any auto-phosphorylated Lck, then the tagged protein was purified by Ni affinity chromatography followed by anion

exchange. The semi-pure protein was treated with TEV protease (~50 μg/mL) overnight at 4°C, then the TEV protease, protease-cleaved His$_6$-tag, and any uncleaved protein were removed by Ni affinity chromatography. The tagless kinase domain was further purified by size exclusion chromatography over a Superdex 200 column.

Constructs for the mouse c-Abl kinase domain (residues 232–502), wild-type and mutant chicken c-Src kinase domains (residues 257–525), human Fyn kinase domain (residues 261–529), and human Hck kinase domain (residues 252–520) all contained an N-terminal His$_6$-tag followed by a TEV protease cleavage site. These proteins were co-expressed in *E. coli* BL21(DE3) cells with YopH phosphatase and purified as described previously for c-Abl and c-Src (*Seeliger et al., 2005*). We note that although the chicken c-Src kinase domain was used for our experiments, all residue numbering in the main text corresponds to human c-Src. The human and chicken c-Src kinase domains only differ at two positions (Thr 357 in human c-Src is a Met residue in chicken c-Src, and Glu 505 in human c-Src is an Asp residue in chicken c-Src). These positions are not near the substrate docking site and are unlikely to impact activity or specificity.

## In vitro measurements of phosphorylation rates with purified kinases and peptides

To validate the specificity trends observed in high-throughput screens, we carried out in vitro measurements of phosphorylation kinetics using purified kinases and synthetic peptides derived from sequences in the Human-pTyr library (purchased from Elim Biopharmaceuticals, Inc., Hayward, CA). These peptides (with sequences given in *Figure 3E*), were capped with an N-terminal acetyl group and a C-terminal carboxamide, and correspond to the following human phosphosites: PDGFRα Tyr 762 Y768A (peptide 1), Tensin-1 Tyr 366 (peptide 2), PKCδ Tyr 313 (peptide 3), nitric oxide-associated protein 1 Tyr 77 (peptide 4), and potassium voltage-gated channel subfamily A member 3 Tyr 187 (peptide 5). Kinetic assays were carried out as described previously, using a continuous colorimetric assay (*Barker et al., 1995*; *Shah et al., 2016*). In each case, the kinase domain concentration was 500 nM, the peptide concentration was 250 μM, the ATP concentration was 100 μM, and the measurements were made at 37°C. Initial velocities were determined by fitting the earliest time points, in the linear regime of the reaction progress curves, to a straight line and extracting the slopes. Measurements were typically carried out in triplicate, and the background ATP hydrolysis rate for each kinase, determined by monitoring a reaction in the absence of a peptide substrate, was subtracted from the rate observed for each kinase-peptide pair.

## Phylogenetic analysis of Src-family kinases

Two phylogenetic analyses were carried out on sequences of Src-family kinases. To generate the phylogenetic tree shown in *Figure 1C*, sequences of the kinase domains of the eight human Src-family kinases and one related non-Src-family kinase, c-Abl, were aligned using T-Coffee (*Notredame et al., 2000*). This alignment was used to infer a rooted phylogenetic tree using Phylip (*Felsenstein, 1989*), with the c-Abl sequence chosen as the outgroup. The c-Abl outgroup branch was removed from the figure for clarity. For the sequence logos shown in *Figure 4—figure supplement 1, a* larger alignment of Src-family kinases was generated using eight Src-family kinases from nine diverse jawed vertebrates, *Homo sapiens*, *Mus musculus*, *Bos taurus*, *Gallus gallus*, *Chelonia mydas*, *Anolis carolinensis*, *Xenopus tropicalis*, *Danio rerio*, and *Callorhinchus milii*, and two Src-family kinases from the jawless vertebrate *Lethenteron camtschaticum*. The sequences were aligned using T-Coffee (*Notredame et al., 2000*), then split into two sub-alignments, one for Src-A kinases and another for Src-B kinases. Regions of interest from these sub-alignments were visualized as sequence logos using WebLogo (*Crooks et al., 2004*).

## Molecular dynamics simulation of substrate-bound Lck

To determine plausible kinase-substrate electrostatic interactions that may differentiate Lck and c-Src, we carried out a 500 ns molecular dynamics simulation of the human Lck kinase domain (residues 231–501) bound to a peptide derived from the T cell receptor ζ-chain (residues 104–118). For this simulation, a crystal structure of the Lck kinase domain in an active conformation, bound to AMP-PNP, a stable analog of ATP, was used to model the starting coordinates of the enzyme (PDB code 1QPC) (*Zhu et al., 1999*). To model the configuration of enzyme and substrate residues in the

active site, a crystal structure of the insulin receptor kinase domain bound to a substrate was used as a guide (PDB code 1IR3) (*Hubbard, 1997*). To construct a starting model, the kinase domains of Lck and the insulin receptor were aligned, and the backbone atoms in the activation loop of Lck were adjusted to match the conformation of the insulin receptor activation loop as seen in PDB entry 1IR3. The coordinates for the ATP analog in the Lck structure were replaced with that of the ATP analog, two $Mg^{2+}$ ions, and six ordered water molecules coordinating the $Mg^{2+}$ ions in the insulin receptor structure. The non-hydrolyzable ATP analog was converted to ATP by changing the appropriate atom names and types in the coordinate file. For the substrate peptides, the coordinates for the entire target tyrosine residue and the backbone atoms for residues +1 to +3 from the insulin receptor structure were appended to the Lck kinase domain coordinates. To complete the substrate-bound Lck model, the remaining substrate atoms were manually built onto the structure, ensuring reasonable backbone dihedral angles, reasonable side chain rotamers, and no steric clashes. Except for the target tyrosine and the +1 to +3 positions, all other residues were modeled projecting away from the kinase domain in an arbitrary conformation. Both polypeptide chains were capped at their N- and C-termini with N-acetyl and N-methylcarboxamide groups, respectively.

The molecular dynamics trajectory was generated using Amber14 (*Case et al., 2014*), with the Amber ff99SB force field (*Lindorff-Larsen et al., 2010*). The TIP3P water model was used (*Jorgensen et al., 1983*), and seven sodium ions were required to neutralize the system. After the initial energy minimization steps, the system was heated to 300 K, followed by four 500 ps equilibration steps at constant number, pressure, and temperature (NPT), with harmonic positional restraints on the protein, peptide, ATP, and $Mg^{2+}$ atoms. Next, the system was subject to another one ns equilibration step at constant number, volume, and temperature, with no positional constraints on any atoms. Finally, production runs were carried out under NPT conditions. Periodic boundary conditions were imposed, and particle-mesh Ewald summations were used for long-range electrostatic calculations (*Darden et al., 1993*). The van der Waals cut-off was set at 10 Å. A time step of 2 fs was employed and the structures were stored every 2 ps.

## Acknowledgements

We thank Hector Nolla and Alma Valeros of the Flow Cytometry Core Facility at UC Berkeley for their assistance with cell sorting; Bill Russ and Rama Ranganathan for their insights and for assistance with deep sequencing; and Aaron Cantor for helpful suggestions throughout the design and implementation of this investigation. This work was supported in part by NIH grant P01 AI091580 to AW and JK. NHS is supported by the Damon Runyon Cancer Research Foundation postdoctoral fellowship. ML is supported by the PROMOS scholarship from the University of Würzburg, provided by the German Academic Exchange Service (DAAD).

## Additional information

### Competing interests

John Kuriyan: Senior editor, *eLife*. The other authors declare that no competing interests exist.

### Funding

| Funder | Grant reference number | Author |
| --- | --- | --- |
| Damon Runyon Cancer Research Foundation | | Neel H Shah |
| German Academic Exchange Service London | | Mark Löbel |
| National Institutes of Health | P01 AI091580 | Arthur Weiss John Kuriyan |

The funders had no role in study design, data collection and interpretation, or the decision to submit the work for publication.

## Author contributions
Neel H Shah, Conceptualization, Resources, Data curation, Formal analysis, Validation, Investigation, Visualization, Methodology, Writing—original draft, Project administration, Writing—review and editing; Mark Löbel, Formal analysis, Investigation, Writing—review and editing; Arthur Weiss, Conceptualization, Supervision, Funding acquisition, Writing—review and editing; John Kuriyan, Conceptualization, Supervision, Funding acquisition, Writing—original draft, Project administration, Writing—review and editing

## Author ORCIDs
Neel H Shah http://orcid.org/0000-0002-1186-0626
Arthur Weiss http://orcid.org/0000-0002-2414-9024
John Kuriyan http://orcid.org/0000-0002-4414-5477

## Decision letter and Author response
Decision letter https://doi.org/10.7554/eLife.35190.024
Author response https://doi.org/10.7554/eLife.35190.025

## Additional files

### Supplementary files
• Source code 1. translateFastq.py A python script to convert a fastq file of DNA sequences derived from a high-throughput bacterial-display screen into a fasta file of the corresponding amino acid sequences. Sequences will be translated according to the first reading frame. Accordingly, the DNA sequences should be trimmed to remove adapter sequences, as described previously (*Shah et al., 2016*).
DOI: https://doi.org/10.7554/eLife.35190.021

• Transparent reporting form
DOI: https://doi.org/10.7554/eLife.35190.022

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
