## [Decision Letter]

Thank you for submitting your article "Fine-tuning of substrate preferences of the Src-family kinase Lck revealed through a high-throughput specificity screen" for consideration by *eLife*. Your article has been reviewed by two peer reviewers, and the evaluation has been overseen by a Reviewing and Senior Editor. The following individual involved in review of your submission has agreed to reveal his identity: Benoît Roux (Reviewer #2).

The reviewers have discussed the reviews with one another and the Reviewing Editor has drafted this decision to help you prepare a revised submission.

Summary:

This manuscript follows on work the authors previously published in *eLife* examining the substrate specificity of non-receptor tyrosine kinases involved in T cell receptor signaling. In the previous paper, the authors developed a bacterial surface display method that leverages next-generation sequencing technology to identify peptides phosphorylated by tyrosine kinases from a pool containing hundreds of sequence variants. Using this method the authors revealed substantial differences between the kinases ZAP-70 and Lck that allowed them to rationalize their sequential activation and specific targeting of ZAP-70 substrates. In the present manuscript, the authors extend their originally reported method by generating libraries consisting of thousands of peptides corresponding to known sites of phosphorylation within the human proteome. After validating the new library using well characterized kinases (Abl and ZAP-70), they go on to analyze Lck and the related kinase Src. As anticipated from previous analyses of the two kinases, their specificity was broadly similar when the preferred sequences are considered in aggregate. What is really interesting about their findings is that there are a substantial number of individual peptide sequences that are reproducibly phosphorylated preferentially by either Src or Lck, a phenomenon that would not have been reasonably predicted based on prior literature. They further note that Src specific substrates tended to have a more negative net charge in comparison to Lck specific substrates. Strikingly, mutation of specific residues near the peptide binding site of Src to decrease charge shifts its specificity to be more like Lck, providing a reasonable model to explain at least some of their relative substrate preferences.

This study is a nice extension of their previous paper, and seems appropriate for a "research advance". The incorporation of large numbers of proteome-derived sequences in particular expands the capabilities of their method. One benefit that is clear from the current study is that by analyzing a large substrate pool one is able to better tease out residues that are negatively selected by the kinase.

Minor points:

1) While the focus on the differences in the electrostatic selectivity of Src and Lck is reasonable and makes sense as an extension of the work on ZAP-70, the logos showing peptide motifs that are differentially phosphorylated by Src and Lck show several statistically significant features that aren't really discussed in the manuscript. These are at key positions at which tyrosine kinases tend to be most selective (-1 and +3). Can these differences be rationalized based on distinct residues in the catalytic clefts of the two kinases?

2) It would be interesting to do some analysis (other than just showing the sequence logos) of peptides that are either commonly or differently phosphorylated by the two kinases. For example, are established phosphorylation sites of the kinases enriched within their selected peptide pools? Another question is whether there might be multiple distinct sequence motifs in the data that are "buried" when examined in aggregate. Another method that similarly analyzed phosphorylation of a large number of individual peptides (Kettenbach et al., 2012) was able to identify multiple sequence motifs using a web-accessible search tool (Motif-X).

3) For ease of comparison, Figure 4B should have an additional graph showing data for WT Src.

4) We recommend including lists of the peptide sequences used in the library (and the proteins from which they are derived) along with the enrichment ratios for each kinase examined.

5) The assay with the kinases (c-Src, Lck, and ZAP-70) are carried out from the extracellular side where the target peptides are exposed. These kinases are soluble constructs, that presumably do not comprise the membrane anchoring region. What are the possible consequences of this, and how should one think about the effect of membrane anchoring?

6) Src-family kinases are believed to become activated through a trans-autophosphorylation of the activation loop (activation loop). Is the sequence of the activation loop consistent with the present findings? Is there some inherent specificity in the trans-autophosphorylation process?

7) Because of the variations in hydrophobic residues, the surface-display levels of each peptide may vary. However, no correction was made for this variability. Could one conceive of a chemical control to assess these differences? For example, could one utilize a chemical or enzymatic tool that modifies exposed Tyr residues without sequence selectivity, and then interrogate the results via mass spectroscopy?

---

## [Author Response]

Minor points:1) While the focus on the differences in the electrostatic selectivity of Src and Lck is reasonable and makes sense as an extension of the work on ZAP-70, the logos showing peptide motifs that are differentially phosphorylated by Src and Lck show several statistically significant features that aren't really discussed in the manuscript. These are at key positions at which tyrosine kinases tend to be most selective (-1 and +3). Can these differences be rationalized based on distinct residues in the catalytic clefts of the two kinases?

The reviewers are correct in noting that there are differences in specificity between each of the kinases analyzed beyond general electrostatic preferences, most notably at the critical -1 and +3 positions in substrates relative to the tyrosine phosphorylation site. In our previous paper, we postulated that a key residue on the F-G loop of tyrosine kinases, Lys 538 in ZAP-70 and Ile 439 in Lck, recognizes the -1 residue on substrates and that this can largely explain the difference between those kinases with respect to -1 residue preferences. This same residue is valine in c-Src, which may explain the subtle difference in hydrophobic residue preferences at the -1 position relative to c-Src. c-Abl has a serine at this position on the F-G loop, and c-Abl is generally more promiscuous with respect to -1 residue preferences.

Based on our molecular dynamics simulations and based structural data in the literature, recognition of the +3 position is mediated primarily by two residues in the catalytic cleft of tyrosine kinases, one immediately downstream of the activation loop and another on the G-helix. The latter has been noted by Till et al. as important for c-Abl recognition of a +3 proline residue (PMID: 9988744). Lck and c-Src also differ at these positions.

We have included a discussion and depiction of these observations in a new section at the end of the ‘Results and Discussion’ section, and in the newly added in Figure 5. These additional materials include a structural rendering of Lck bound to a peptide, highlighting the three critical positions on the kinase domain, a summary of the residue identities at those positions in all four kinases analyzed in this manuscript, and a simplified rendering of the -1 and +3 preferences for each kinase, derived from our high-throughput screening data. We have also included experimental data in Figure 5—figure supplement 1 showing that mutation of the aforementioned F-G loop residue in c-Src alters its -1 preference.

2) It would be interesting to do some analysis (other than just showing the sequence logos) of peptides that are either commonly or differently phosphorylated by the two kinases. For example, are established phosphorylation sites of the kinases enriched within their selected peptide pools? Another question is whether there might be multiple distinct sequence motifs in the data that are "buried" when examined in aggregate. Another method that similarly analyzed phosphorylation of a large number of individual peptides (Kettenbach et al., 2012) was able to identify multiple sequence motifs using a web-accessible search tool (Motif-X).

We agree with the reviewers that a deeper analysis of the large datasets produced by our screens of the Human-pTyr Library could yield additional insights into kinase specificity. We have included the results of two analyses in line with the reviewers’ suggestions in a new section at the end of the ‘Results and Discussion’ entitled ‘Additional insights from specificity screens with the Human-pTyr library’.

First, we analyzed the pool of efficiently phosphorylated peptides for each kinase using Motif-X (PMID: 16273072) to identify distinct sequence motifs that might be buried in the full sequence logos. For c-Src and Lck, this analysis recapitulated the conserved -1 Ile, Val, Leu, and Thr preferences for each kinase, but also revealed a statistically significant +3 Phe motif for c-Src that is absent in Lck. For c-Abl and ZAP-70, this analysis showed that multiple positions (such as -1, +1 and +3) independently contribute to substrate recognition. For example, either a -1 Ile or +3 Pro are sufficient to yield an efficiently phosphorylated c-Abl substrate. These results are summarized in the text and in Figure 5—figure supplement 2.

Second, we compared a curated list of reported human kinase-substrate pairs in the PhosphoSitePlus database with our screening results. For c-Abl, ZAP-70, c-Src, and Lck, approximately 40-60% of the reported substrates in the database are efficiently phosphorylated in the surface-display screen, as depicted in Figure 5C and enumerated in Figure 5—source data 1. Although all of the kinase-substrate pairs in this database may not be unambiguously confirmed, this modest correlation between the database and our screens raises two important points about tyrosine kinase substrate selection. (1) While the local sequences surrounding tyrosine phosphosites likely play a significant role in substrate selection by tyrosine kinases, other factors such as localization and recruitment are also important and may partially override sequence preferences. (2) Tyrosine kinase-substrate interactions may not necessarily be optimized for the most efficient phosphorylation. There may be situations where sub-optimal phosphorylation kinetics are required in a signaling pathway, and so natural substrates deviate from idealized sequence motifs.

3) For ease of comparison, Figure 4B should have an additional graph showing data for WT Src.

The data in Figure 4B represent the residue abundances in the pool of peptides that are efficiently phosphorylated by each c-Src mutant but not by wild-type c-Src. For each mutant, we have added an additional graph below the original graph that shows the residue abundances in the pool of peptides phosphorylated by wild-type c-Src but not by the mutant. In addition, we have changed the figure labels on these histograms to more clearly indicate which peptides are being represented in each graph.

4) We recommend including lists of the peptide sequences used in the library (and the proteins from which they are derived) along with the enrichment ratios for each kinase examined.

We have appended a table to the revised manuscript that contains all of the peptide sequences, the proteins from which they were derived, the tyrosine phosphosite residue number in that protein, and the enrichment scores for c-Src (wild-type and mutants), Lck, c-Abl, and ZAP-70. These data can be found in Figure 2—source data 1.

5) The assay with the kinases (c-Src, Lck, and ZAP-70) are carried out from the extracellular side where the target peptides are exposed. These kinases are soluble constructs, that presumably do not comprise the membrane anchoring region. What are the possible consequences of this, and how should one think about the effect of membrane anchoring?

The reviewers are correct in noting that the kinase constructs used in this study do not contain a membrane anchoring region. The major impact of membrane anchoring on kinase specificity would be to localize the kinase in proximity to its substrates, and possibly away from proteins it should not phosphorylate. Undoubtedly, sub-cellular localization of enzymes and substrates plays an important role in kinase signaling. The dominant effect of localization on substrate recognition is likely one major reason why only half of the reported substrates of these kinases are high-efficiency substrates in the bacterial surface-display screens. Another interesting consideration is that anchoring the kinase at the membrane can impose geometric constraints on substrate recognition. For example, given that many phosphotyrosine signaling proteins form dense clusters at the membrane, the relative orientations of kinases and their substrates may be constrained in a way that dictates specificity. Given that the role of localization in kinase signaling is described in the introduction, we felt that this issue did not require further modification of the manuscript.

6) Src-family kinases are believed to become activated through a trans-autophosphorylation of the activation loop (activation loop). Is the sequence of the activation loop consistent with the present findings? Is there some inherent specificity in the trans-autophosphorylation process?

Not all of the Src-family and Abl-family activation loop sequences are well-represented in the Human-pTyr Library, most likely due to poor PCR amplification of the coding sequence during library construction (see Materials and methods). The ZAP-70 activation loop sequence and that derived from the Src-family kinase Blk are abundant in the library. Analysis of these peptides, and comparison of activation loop sequences to the motifs derived from our screens, suggests that activation loop sequences of tyrosine kinases are not optimized for auto-phosphorylation. This is consistent with the observation in our previous paper that tyrosine kinases must make specific contacts between the C-lobes of their kinase domains for efficient *trans*-autophosphorylation of their activation loops. Notably, in our screens, Lck can robustly phosphorylate Tyr 493 in ZAP-70 activation loop, consistent with its activity in T cell signaling. It is also noteworthy that, in our screens, the Src-family kinases are poor enzymes for phosphorylation of their negative regulatory C-terminal tail sequences, consistent with the requirement for Csk to phosphorylate those tails. The structural basis for activation loop autophosphorylation in Src-family kinases was a major focus of our previous paper, and so we do not discuss this explicitly in this manuscript.

7) Because of the variations in hydrophobic residues, the surface-display levels of each peptide may vary. However, no correction was made for this variability. Could one conceive of a chemical control to assess these differences? For example, could one utilize a chemical or enzymatic tool that modifies exposed Tyr residues without sequence selectivity, and then interrogate the results via mass spectroscopy?

As noted by the reviewers, we did not correct for variation in the surface-display levels of peptides in the Human-pTyr Library. There may be efficient chemical labeling and mass spectrometric methods to examine surface-display levels. We previously reported a method, analogous to our measurements of phosphorylation efficiency, that utilizes cell sorting and deep sequencing to accurately determine relative surface-display levels. Our surface-display scaffold bears an epitope tag lacking tyrosine residues that is distinct from the displayed substrate peptide. This tag can be detected with a fluorescently-labeled antibody and used to separate cells by fluorescence-activated cell sorting on the basis of surface-display level. By collecting cells from multiple bins spanning the surface-display level distribution, and deep sequencing cells within each bin, one can quantitatively determine the relative surface-display levels of all peptides in the library.

In our previous study, we described this procedure extensively in an appendix and showed that a correction based on these measurements only marginally impacts the enrichment scores that report on phosphorylation efficiencies. Furthermore, we find that surface-display levels are highly reproducible between screens. As described in the ‘Results and Discussion’, we omitted this correction for the study presented here because most of our analyses involved comparing pairs of kinases against the same library, where any artifacts due to surface-display variation would be the same for each kinase. We have added a sentence to the ‘Results and Discussion’ briefly explaining the methodology that we previously developed to compare surface-display levels, and we refer the readers to our earlier report for a thorough description of this procedure.